# Screen-Shooting Resilient Watermarking Scheme via Learned Invariant Keypoints and QT

**DOI:** 10.3390/s21196554

**Published:** 2021-09-30

**Authors:** Li Li, Rui Bai, Shanqing Zhang, Chin-Chen Chang, Mengtao Shi

**Affiliations:** 1School of Computer Science and Technology, Hangzhou Dianzi University, Hangzhou 310018, China; lili2008@hdu.edu.cn (L.L.); bairui@hdu.edu.cn (R.B.); sqzhang@hdu.edu.cn (S.Z.); shimt@hdu.edu.cn (M.S.); 2Key Laboratory of Brain Machine Collaborative Intelligence of Zhejiang Province, Hangzhou 310018, China; 3Department of Information Engineering and Computer Science, Feng Chia University, Taichung 40724, Taiwan

**Keywords:** screen-shooting, FRFS, QT, robustness, partial shooting

## Abstract

This paper proposes a screen-shooting resilient watermarking scheme via learned invariant keypoints and QT; that is, if the watermarked image is displayed on the screen and captured by a camera, the watermark can be still extracted from the photo. A screen-shooting resilient watermarking algorithm should meet the following two basic requirements: robust keypoints and a robust watermark algorithm. In our case, we embedded watermarks by combining the feature region filtering model to SuperPoint (FRFS) neural networks, quaternion discrete Fourier transform (QDFT), and tensor decomposition (TD). First we applied FRFS to locate the embedding feature regions which are decided by the keypoints that survive screen-shooting. Second, we structured watermark embedding regions centered at keypoints. Third, the watermarks were embedded by the QDFT and TD (QT) algorithm, which is robust for capturing process attacks. In a partial shooting scenario, the watermark is repeatedly embedded into different regions in an image to enhance robustness. Finally, we extracted the watermarks from at least one region at the extraction stage. The experimental results showed that the proposed scheme is very robust for camera shooting (including partial shooting) different shooting scenarios, and special attacks. Moreover, the efficient mechanism of screen-shooting resilient watermarking could have propietary protection and leak tracing applications.

## 1. Introduction

Taking photographs has become highly efficient and convenient as shown by the widespread use of smart phone, pinhole, augmented reality (AR) and virtual reality (VR) cameras as well as mini digital video recorders (DVRs). However, this new efficiency could pose a threat to information security field. Capturing computer screen photos and videos is now an important means of stealing internal confidential information, which is difficult to prohibit and leaves no trace. Therefore, a robust watermarking scheme that can extract information from screen-shot photos should be designed to protect confidential information. We can embed identifying information, such as a screen or user identifer number, in the host image, and through the extracted message, we can provide a reliable way to authenticate an images and protect copyright. Figure 1 shows a diagram of screen-shooting watermarking.

Various methods have been proposed for image watermarking that are mostly robust to conventional image attacks [1,2,3] but are vulnerable to multiple attacks. These schemes are not typically designed to work for screen-capture photos, but in recent years, print-to-scan [4,5], print-to-capture [6,7], and screenshot [8] scenarios have been studied extensively. However, screen-shooting requires more sophisticated and new attacks, such as moiré, different scale display, lens distortion, and light source distortion. When taking photos from a screen, the images and watermarks undergo a suite of A-to-D and D-to-A processes that can be regarded as hybrid strong attacks [3]. In addition, screen-shot photos also suffer from high-scale compression caused by social media platforms, such as uploading to WeChat, so partial shooting is restricted to specific shooting scenarios [3,6,9].

Many screen-shooting watermarking schemes have been proposed in recent years [9,10,11,12,13,14,15,16,17], and they can be classified into two types.

The first is anti-screen shooting based on image coding. These methods use visual characteristics and combine software and hardware to change the brightness of a special display pattern superimposed on the host image or a screen. Cui et al. [11] designed a cross-component correlation extraction method to detect image barcode from color images. Nakamura et al. [12] proposed a scheme based on an additive template that was added to the host image to embed a watermark. Gugelmann et al. [10] developed a coding scheme on the strength of convolutional codes that complements the watermarking and solves the special requirements of screen watermarking. In [15], a method was proposed for embedding a watermark in print media that could blindly detect a watermark by using a mobile phone. In [16], a parametric print-to-capture channel model and an efficient channel estimation scheme requiring low training overhead were proposed. Nevertheless, the aforementioned methods severely affect the visual quality of the images.

The second type is based on localization preprocessing and frequency-domain transformation. The fusion of feature points and digital watermarking technology in the frequency domain could effectively solve the special attack of screen-shooting. Most existing screen-shooting methods carry out perspective correction and resizing to yield an undistorted version of the original image. This study combines keypoint detection of deep learning with double-transformation watermarking in the frequency domain. Our scheme can achieve blind extraction of watermarks and detection of keypoints and does not require perspective correction.

The main contributions of this work can be summarized as follows:We apply a modified version of the keypoint detector SuperPoint. Specifically, we add a new model called Feature Regions Filtering model to SuperPoint (FRFS).We propose a screen-shooting watermarking scheme via learned invariant keypoints, which combines FRFS, QDFT, and TD (FRFSQT).The proposed scheme makes the most of the merits of a traditional watermarking algorithms and deep learning neural networks to build an efficient mechanism to protect proprietary information that is resilient to screen-shooting attacks.

## 2. Related Work

In this section, we review keypoint detection, watermark algorithms, SuperPoint, TD, QDFT, and screen-shooting attacks.

### 2.1. Local Feature Keypoint Detection

Local feature keypoints have been widely employed in robust image watermarking for localization preprocessing. Existing keypoint detection methods mainly include the following types: SIFT [18,19,20], SURF [21,22], Harris [23,24], BRISK [25], FAST [26], BRIEF [27] and ORB [28]. Over the past few years, keypoint detection methods based on the applicability and potential of machine learning, particularly deep learning, have superseded these traditional approaches [29,30,31,32,33,34,35,36,37]. Yi et al. [29] proposed a deep-learning framework based on a conventional neural network to detect feature points, direction estimation, and descriptor extraction. Verdie et al. [30] proposed a keypoint detection algorithm based on learning that can deal well with the changes of different scenes. Daniel et al. [37] proposed a FCNN model for keypoint detection and descriptor generation based on SuperPoint self-supervised optimization. Liu et al. [35] introduced a scheme descriptor generation dubbed GIFT based on group of transformations network. Yuki et al. [36] introduced a novel deep model that learns local features by a local feature pipeline and does not require human supervision.

### 2.2. Watermarking Algorithm in Frequency Domain

Frequency-domain watermarking technology, such as DWT [38], DFT [39], DCT [40], QDFT [41,42] and tensor decomposition [15,43,44], helps improve imperceptibility and robustness. For instance, Fang et al. [9] proposed an intensity-based SIFT algorithm to extract complete watermark information from the screen image to protect confidential information in the DCT domain. However, SIFT keypoints are not consistent and stable under screen-shooting attacks. Lorenzo et al. [45] improved a strategy for watermarking on color images by using a mobile phone based on the Neyman–Pearson criterion. Fang et al. [17] proposed a screen-to-camera image code named “TERA”, which can be widely used in many applications for copyright protection by using a leak-tracing watermark. Fang et al. [46] designed a novel document underpainting watermarking algorithm resilient to camera-shooting.

### 2.3. SuperPoint

SuperPoint is a FCNN model for keypoint detection and descriptor generation proposed in 2018 by MagicLeap that uses a self-supervised domain-optimized framework [37].

The input of SuperPoint is an image, and the output is a heatmap of the same size. Figure 2 shows the flowchart of a keypoint detector. The model uses a single and shared encoder to process an RGB image into a gray image to reduce input image dimensionality. Through the encoder processing, the model uses two decoders for keypoint detection and descriptor generation. Most of the network’s parameters are shared between the keypoint detection and descriptor generation, which depends on the architecture.

In this work, we adapt the keypoint detector of the SuperPoint neural network. The detector is used to locate the embedded regions in the shooting picture. The locating algorithm achieves blind extraction and requires no prior information.

### 2.4. QDFT and TD Watermarking Algorithm

In this subsection, we introduce the double-transformation watermarking algorithm based on TD and QDFT in the frequency domain.

#### 2.4.1. QDFT

We take the description of QDFT from Sangwine [47]. Considering that it does not satisfy the commutative law, QDFT is divided into three types: left-way transform $F_{L}$, right-way transform $F_{R}$ [48], and hybrid transform $F_{LR}$ [47]. The form of the left-way transform $F_{L}\left(\lambda ,\upsilon \right)$ is
(1)$$F_{L}\left(\lambda ,\upsilon \right)=\frac{1}{\sqrt{X,Y}}\sum_{x=0}^{X-1}\sum_{y=0}^{Y-1}e^{-\theta 2\pi (\frac{x\lambda }{X}+\frac{y\upsilon }{Y})}f\left(x,y\right).$$

Color image pixels have three components: R, G, and B. Thus, they can be represented in a quaternion form by using a pure quaternion. For example, the coordinates of a pixel are (x,y) in a color-image can be represented as follows:(2)f(x,y)=R(x,y)i+G(x,y)j+B(x,y)k,
where R(x,y) is the red component; G(x,y) is the green component; and B(x,y) is the blue component of a color image. f(x,y) is a color image of size X×Y represented in the quaternion form as Equation (Equation 3). The inverse QDFT [48] is defined by
(3)$$f\left(x,y\right)=\frac{1}{\sqrt{X,Y}}\sum_{x=0}^{X-1}\sum_{y=0}^{Y-1}e^{\theta 2\pi (\frac{x\lambda }{X}+\frac{y\upsilon }{Y})}F_{L}\left(\lambda ,\upsilon \right).$$

In these definitions, the quaternion operator is generalized, and θ can be any unit of a pure quaternion, where $\theta^{2}=-1$. The operators *i*, *j*, and *k* are special cases of θ; in this paper we take, $\theta =\left(i+j+k\right)/\sqrt{3}.$

Using Equations (1) and (2), we can obtain A(λ,υ), the real component, and C(λ,υ), D(λ,υ), and E(λ,υ), the three imaginary components in Equation (Equation 4).
(4)$$F_{L}\left(\lambda ,\upsilon \right)=A\left(\lambda ,\upsilon \right)+C\left(\lambda ,\upsilon \right)i+D\left(\lambda ,\upsilon \right)j+E\left(\lambda ,\upsilon \right)k.$$

The three imaginary components *C*, *D*, and *E* also have a strong correlation. Hence, the three components can be used to construct a tensor *T*.

#### 2.4.2. TD

TD is an efficient technique used in many fields. Two particular tensor decompositions are considered higher-order extensions of the matrix singular value decomposition: CANDECOMP/PARAFAC (CP) and Tucker decomposition, which is always selected to implement TD. A third-order tensor $T\in R^{M\times N\times O}$ is decomposed by the Tucker decomposition to three orthogonal factor matrices $U^{1}\in R^{M\times P}$, $U^{2}\in R^{N\times Q}$, $U^{3}\in R^{O\times R}$, and a core tensor $K\in R^{P\times Q\times R}$ [49].

Each element in the core tensor *K* represents the degree of interaction among different slices. The Tucker decomposition [50] is defined in Equation (Equation 5) as
(5)$$T\approx K\times_{1}U^{1}\times_{2}U^{2}\times_{3}U^{3}\approx \left[\left[K;U^{1},U^{2},U^{3}\right]\right].$$
and for each element of the original tensor *T*, it [50] is expressed in Equation (Equation 6).
(6)$$T\approx \sum_{p=1}^{P}\sum_{q=1}^{Q}\sum_{r=1}^{R}k_{pqr}u_{p}^{1}\circ u_{q}^{2}\circ u_{r}^{3},$$
where *P*, *Q*, and *R* correspond to the numbers of column vectors of the factor matrices $U^{1}$, $U^{2}$, and $U^{3}$, respectively. *P*, *Q*, and *R* are generally less than or equal to *M*, *N*, and *O*, respectively. The symbol ‘∘’ represents the outer product between two vectors. The symbol ‘[[ ]]’ is a concise representation of Tucker decomposition given in [50]. The core tensor *K* has the same dimension as tensor *T*, and it is expressed in Equation (Equation 7).
(7)$$K\approx T\times_{1}U^{1}\times_{2}U^{2}\times_{3}U^{3}.$$
*K* has full orthogonality; that is, any two slices of the core tensor *K* are orthogonal to each other, and the inner product between the two slices is zero.

### 2.5. Screen-Shooting Attacks

The screen-shooting process can be regarded as a cross-media information transfer process. It contains a special attack and other traditional distortions, such as scale, translation, and rotation and image processing. Kim et al. [19] summarized the distortion of the screen-shooting process into four attacks: display, lens, sensor, and process. Among them, Fang et al. [9] also proposed three categories: lens deformation, light source deformation, and moiré pattern distortion. In Figure 3, we summarized the four distortions of screen-shooting.

## 3. Proposed Scheme

This study focused on image watermarking schemes in the invariant domain and learned invariant keypoints. Specifically, we combined FRFS, QDFT, and TD. The proposed scheme made the most of the merits of a traditional watermarking algorithms and deep-learning neural networks to create an efficient mechanism for screen-shooting scenarios. Next, we elaborated the embedding and watermark extraction procedures in Section 3.2 and Section 3.3, respectively. We also formulated the frameworks of embedding and extraction in Figure 4 and Figure 5, respectively. We describe the optimized FRFS model in Section 3.1.

### 3.1. Feature Region Filtering Model

To meet our demand and improve SuperPoint’s performance, we developed SuperPoint neural networks. We applied and modified the keypoint detector of SuperPoint and added a new model called Feature Regions Filtering to Superpoint (FRFS) to select non-overlapping embedding regions. Point detection heatmaps (SuperPoint’s outputs) were generated by keypoint confidence. The keypoint detector’s loss function $L_{p}$ was a fully convolutional cross-entropy loss, and the specific procedure can be found in [37].

Considering that the embedding regions centered at each keypoint should be sifted, the operation can be regarded as the following formulation, which can be solved using Equations (8) and (9) called “Feature Regions Filtering (FRF)”. When the keypoint confidence *h* > threshold, we obtain $a_{k}$ points and k,g∈[1,640×480].
(8)$$R\left(a_{k}\right)\cap R\left(a_{g}\right)=⌀\;\left(k\neq g\right)\;,$$
Here $R(a_{k})$ are the regions of size 32×32, which should be disjointed. If Equation (Equation 8) is workable, the two regions are saved; however, if $R\left(a_{k}\right)\cap R\left(a_{g}\right)\neq ⌀$, and $S(a_{k})$ > $S(a_{g})$, the region $R(a_{g})$ is deleted.
(9)$$DESC\left(\sum_{m=1}^{M}S\left(a_{m}\right)\right)\;\left(m\in \left[1,M\right]\right),$$
where $S(a_{k})$ denotes the strength of the keypoints $a_{k}$; $R(a_{k})$, $R(a_{g})$, and $R(a_{m})$ denote the embedding feature regions centered at $a_{k}$, $a_{g}$, and $a_{m}$, respectively; *M* denotes the non-overlap regions number of an image; DESC denotes descending order; and the watermark capability is 16×M.

High confidence causes clustering, which prevents choosing additional feature regions. First, we needed to use FRFS to filter out the keypoints that overlapped the feature regions centered at each keypoint. Then, we sorted the keypoints in descending order of confidence and chose *K* keypoints with non-overlapping regions. Lastly, we chose high-confidence keyponts as the center of feature regions and obtained non-overlapping feature regions.

### 3.2. Embedding Procedure

The process of embedding watermark information:

Step1: Generate a gray image $I_{o}$ from RGB image $I_{rgb}$, and then resize $I_{o}$ and $I_{rgb}$ to obtain $I_{o}^{\prime }$, $I_{rgb}^{\prime }$ of size 480×640.

Step2: Feed $I_{o}^{\prime }$ into FRFS, and the output of FRFS is heatmap $I_{h}$ with the same size as $I_{o}^{\prime }$.

Step3: Locate high confidence keypoints $I_{h}$ and obtain the coordinate set $S_{e}$ of keypoints.

Step4: Map the coordinates of keypoints to the RGB image $I_{rgb}^{\prime }$, and then construct feature regions of size 32×32 centered at each keypoint in image $I_{rgb}^{\prime }$.

Step5: Divide each feature regions into a cell of size 2×2 and apply the QDFT and TD (QT) watermarking algorithm to each feature cell.

The main target of the proposed QDFT and TD watermarking scheme in [51] is a normal image attack scenario. Our watermarking scheme turned out to be robust to screen-shooting, such that we applied the scheme in this work. The hybrid QDFT and TD transform provided better performance than a single transform, had better fidelity, and had more appropriate color images.

QDFT can process the three channels of a color image as a whole instead of as independent channel, so the inherent correlation of the three channels was used to resist distortions.

The well-known Tucker decomposition is always selected to implement TD because it can maintain the internal structure of an image. It was used to obtain the core tensor, which represents the main properties of each slice of the original tensor and reflects the correlation among the slices. The core tensor K is a compressed version of the original tensor *T*. The Tucker decomposition preserves the inherent correlations of RGB three channels, which brings strong robustness to various attacks; accordingly, it enhances the robustness for watermarking.

The hybrid transform allows the watermark energy to propagate synchronously to the RGB three channels rather than one channel. Hence, the robustness of the watermarking scheme can be greatly improved, and higher-precision color image information can be maintained.

Step6: Use the odd–even quantization embedding technique to embed a bit watermark in a core tensor K(1,1,1). Then, obtain watermarked feature region $R{\left(a_{k}\right)}^{\prime }$. The embedding rule is defined as follows:

If K(1,1,1)>0, $\eta =round(K\left(1,1,1\right)/S_{e}$,
(10)$$K\left(1,1,1\right)=\left\{\begin{array}{ll} \; & K(1,1,1)\;\;\;\;if\;w\neq \;\mathrm{mod}\;(\eta ,2),\\ \; & \eta \times S+0.8\times S\;if\;w=\;\mathrm{mod}\;(\eta ,2); \end{array}\right.$$
else K(1,1,1)=−1×K(1,1,1), η=round(K(1,1,1)/S),
(11)$$K\left(1,1,1\right)=\left\{\begin{array}{cc} \; & -K(1,1,1)\;\;\;if\;w\neq \;\mathrm{mod}\;(\eta ,2),\\ \; & -(\eta \times S-0.8\times S)\;if\;w=\;\mathrm{mod}\;(\eta ,2). \end{array}\right.$$

Here *S* is the quantization step; that is, the watermark-embedding strength; round(*) is the rounding operation; and mod(*) is the modulo operation.

Step7: Embed the complete watermark repeatedly in multiple regions, and then obtain watermarked image $I_{wm}^{o}$.

### 3.3. Extraction Process

The process of extracting a watermark is as follows:

Step1: Generate a gray image $I_{wm}$ from RGB image $I_{rgb}^{w}$ and then resize $I_{wm}$ and $I_{rgb}^{w}$ to obtain $I_{wm^{\prime }}$, $I_{rgb}^{w^{\prime }}$ with a size of 480×640.

Step2: Input $I_{wm}^{\prime }$ into RFRS, and the output of RFRS is heatmap $I_{h}^{w}$ with the same size as $I_{wm}^{\prime }$.

Step3: Locate the keypoints of $I_{h}^{w}$ with high confidence and genearate the coordinate set $S_{w}$ of appropriate keypoints.

Step4: Map the coordinates of keypoints to $I_{rgb}^{w^{\prime }}$, then construct feature regions with a size of 32×32 centered at the keypoints in image $I_{rgb}^{w^{\prime }}$.

Step5: Divide each feature regions into a cell with a size of 2×2 and apply the QDFT and TD (namely, QT) watermarking algorithm to each feature cell.

Step6: Use the odd–even quantization technique to extract a bit watermark in position $K_{w}\left(1,1,1\right)$ of each core tensor. The specific extraction rules are as follows:

$K_{w}\left(1,1,1\right)$=$\left|(K_{w}\left(1,1,1\right)\right|$, $\eta =round(K_{w}\left(1,1,1\right)/S)$,
(12)$$K_{w}\left(1,1,1\right)=\left\{\begin{array}{cc} \; & w=1\;if\;\;\mathrm{mod}\;(\eta ,2)=0,\\ \; & w=0\;if\;\;\mathrm{mod}\;(\eta ,2)=1, \end{array}\right.$$
where ‘|*|’ is the functions abs.

Step7: Obtain the complete watermark $w_{e}$ of each feature region through the odd–even quantization rule.

## 4. Experimental Results and Analysis

Camera-shooting is an air wireless channel information diversion that causes distortions, such as moirés, illumination deformation. To resist screen-shooting process, the robustness of the watermarking algorithm and the locating performance of the detector are the core issues. The specific details are illustrated by the following experiments.

To illustrate the performance of the watermarking algorithm, this study used the peak signal to noise ratio (PSNR) [48], and a normalized correlation coefficient (NC) [51,52] to evaluate the visibility and robustness of the watermarking scheme. PSNR was used to describe the fidelity performance, and NC was used to describe the watermarking robustness.

### 4.1. Choosing the Watermark Strength of the Watermarking Algorithm

To balance the robustness and fidelity, this part discusses the embedding strength *S*. We randomly selected five 640×480 images from MS-COCO 2014 to embed watermark. Figure 6 shows the PSNR and NC of the five watermarked images without attack. We set the watermark embedding strength to S∈(10,200). As the value of *S* increased, so did NC, but PSNR decreased. This finding indicated that the robustness of the watermark improved, whereas the image quality deteriorated. When the value of *S* reached 100, NC was close to 1, and the watermark could be completely extracted without being attacked. From Table 1, when *S* = 120, the watermark could be seen by several college students. To balance robustness and fidelity, *S* = 100 and PSNR > 50.

Figure 7 shows the NC after several shooting attacks, consisting of “blur”, “lens”, “illumination”, “moiré”, and “JPEG”. When the images are under attack, the watermark can be extracted by our algorithm, and the NC converges down toward 1.

Our algorithm obtained excellent robustness against screen-shooting attacks. The specific experimental results are indicated in Section 4.2.

### 4.2. Robustness of the Watermarking Algorithm

The hybrid transform watermarking scheme achieved enhanced robustness and fidelity. The scheme effectively considered the overall characteristics of the color images and propagated the watermark information to the three color channels through QDFT and TD.

#### 4.2.1. Proving the Robustness of the Watermarking Algorithm

We conducted numerous tests, and the results indicated that the algorithm in the tensor domain was robust under attack. K(1,1,1) does not vary, when the 10 images were under attacks (e.g., blur, JPEG, lens, and rotation). By contrast, moiré, scaling, and illumination can change the value of K(1,1,1), but the watermark can still be extracted using our algorithm, given that the odevity of K(1,1,1) is not invariant. We realized enhanced robustness from our algorithm.

Figure 8 shows the value of core tensor K(1,1,1) resistance to different attacks. It can be shown that our watermarking algorithm is robust to screen-shooting attacks.

#### 4.2.2. Performance of the Robust Watermarking Algorithm

To illustrate robustness to screen-shooting attacks, corresponding experiments were conducted. Figure 9 provides the PSNR values and NC for the five different attacks. The PSNR values of all watermarked images were adjusted to more than 50 dB by adjusting the embedding strength; meanwhile, we found that the NC was close 1 under attack for watermarked images. The proposed FRFSQT also coped with partial shooting problems. We embedded the same watermark in numerous feature regions to ensure that at least one piece of complete watermark information survived shooting distortion without requiring a whole map. Figure 10 shows the performance for extracting a watermark from a partial shooting photo.

As stated above, we achieved enhanced performance for watermark imperceptibility and robustness.

### 4.3. Fidelity of the Watermarking Algorithm

We designed a user study with 25 students to measure the fidelity of watermarks at different strengths. We embedded watermarks of different strengths into the host image, which was displayed on an “AOC 27G2” 27-inch screen with a resolution of 1920 × 1080 pixels. The mobile phone we used was an iPhone 8 Plus. The participants were told that it was a watermark test, but they were not told its specific location or shape. They were asked to scan the image and describe in detail the watermark they saw. The results are shown in Table 1.

When the watermark strength was greater than 120, some students identified them. When the strength reached 180, all 25 students could. Watermark strength less than 100 was not identified by any participant.

### 4.4. SuperPoint Heatmap

For keypoint detection, each pixel of the output corresponded to a keypoint probability (i.e., detection confidence). A jet colormap was used for visualization, and when the detection confidence was close to 0, the color was a darker blue. Then, sparse feature points were obtained through non-maximum suppression. For each image, the SuperPoint output was a probability heatmap. Figure 16 shows the heatmap results of the detector.

In Figure 11, the first row identifies the original experimental images, and the second row displays the heatmap in according with the jet colormap. The value at each pixel is the probability that this point acts as a keypoint. The larger the value, the higher the probability, and the redder the heatmap point. Subsequently, embedded multiple regions centered at the keypoints.

### 4.5. Estimated Keypoint Correspondences Detected by SuperPoint

This study used SuperPoint’s detector, which is suitable for a large number of multiple-view screen-shooting problems. In this subsection, we provided some quantitative results of the detector for the evaluation of keypoints under screen-shooting attacks. In Figure 12, Figure 13, Figure 14, Figure 15, Figure 16 and Figure 17, the first row on the left side is the original image, and the one the right is the attacked image; the second row on the left is the original image with detection points and on the right is the attacked image with detection points; the third row refers to correct correspondences after the attacks. After robust and repeatable keypoints were detected, a gift descriptor vector was attached to each point for image matching.

This experiment showed that SuperPoint can detect robust keypoints for screen-shooting attacks. We also detected the same points after screen shooting and blindly extracted watermarks from the shooting photo. The experimental results proved that the keypoints are robust to blurring, lens, illumination, moiré, and JPEG compression attacks. Then, we conducted a series of experiments to prove that the keypoints were also robust to different environments.

Figure 18 shows the examples of recaptured photos in different scenarios. Rows 1–3 show photos in distances of 10, 80, and 100 cm. Rows 4–6 show different horizotal perspective angles of $Left_{30}$ and $Right_{20}$. Row 5 shows a photo under blended attacks of moiré, $up_{10}$, and $Right_{15}$. Row 7 shows a photo under a vertical perspective angle of $up_{80}$.

From the last column, we see that, in different scenarios, the keypoints can be matched by their descriptors. The resulting detection is repeatable by SuperPoint’s detector, and repeatable keypoints are often evaluated by matching purpose. In turn, the watermarks are extracted according to the matching keypoints in a multiple-view screen-shooting.

### 4.6. Comparison with Other Papers

Our scheme can be compared with methods in [9,11,16,17,46] to verify its performance as shown in Table 2.

We found that none of existing methods could cope with the partial shooting of an image; that is, the captured photo had to be complete. Most methods carry out perspective correction and also need to resize the same size of the original image to yield a distorted image. In contrast, our scheme achieved blind watermark extraction and detected keypoints without perspective correction or full-map. Moreover, the algorithm repeatedly embedded complete watermark information in an image to ensure that at least one complete watermark survived distortion. Hence, when the shooting photo is part of an image, we extracted the complete watermark.

In sum, it means that our scheme had better performance and was the most robust in all test scenarios.

## 5. Conclusions

A screen-shooting-resilient watermarking algorithm should meet the following two basic requirements: a robust watermark algorithm and robust keypoints. We proposed a novel screen-shooting watermarking scheme via learned invariant keypoints that combined FRFS, QDFT, and TD (namely, FRFSQT) to protect confidential information displayed on a screen.

We analyzed the robust algorithm against screen-shooting attacks and scaling within a reasonable range. The original image was a 640×480, screen capture, and the captured image was 4032 × 3024. As we saw, the captured image was approximately six times larger, so we reduced it six times. Beyond that, the proposed watermarking algorithm was a small size for accuracy of extraction, and we embedded complete watermark information repeatedly to ensure that at least one watermark survived distortion.

The keypoints detected were the most robust and repeatable. When only a part of the protected image was captured, we also used FRFS to filter out the keypoints. Specifically, we sorted them by confidence and retained the higher confidence keypoints because these were likely to be the robust keypoints for tracing applications. For high-confidence cause clustering, we used Feature Regions Filtering to remove overlaps. If many new feature points were detected, we also constructed feature regions centered on the new keypoints. But if the locating was not accurate, the extracted watermark bits were relatively random; therefore, the similarity between the two watermarks was very small.

The proposed scheme made the most of the merits of traditional watermarking algorithms and a deep-learning neural networks to establish an efficient mechanism that protects proprietary information by being resilient to screen-shooting attacks. Moreover, keypoint detection by SuperPoint had greater repeatability. Our results proved that our scheme exceeded state-of-the-art methods. Compared with previous schemes, ours provided remarkable improvement in extraction potency and robustness for screen-shooting process.

To improve the performance of the mechanism, we hope to reduce the time complexity and design more watermarking algorithms with higher robustness. 

## Figures and Tables

**Figure 1 sensors-21-06554-f001:**
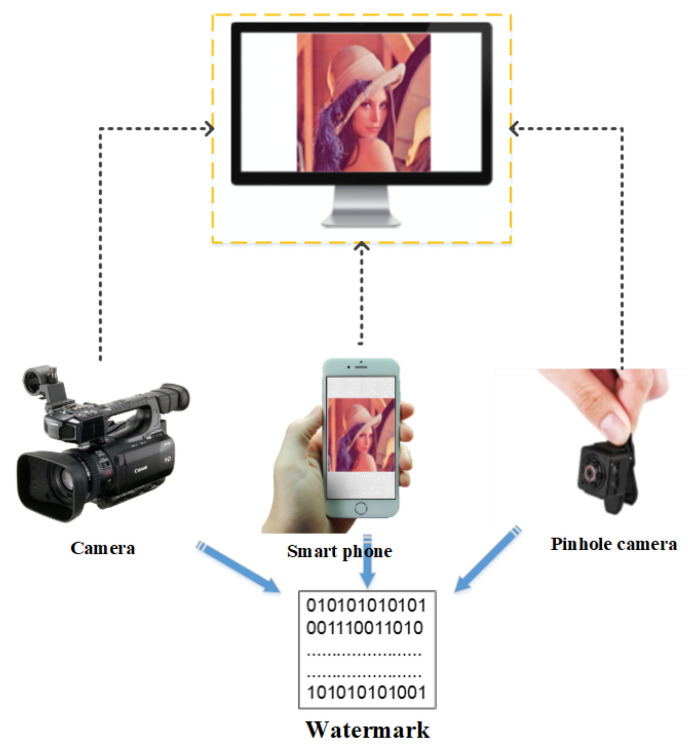
Schematic diagram of screen shooting.

**Figure 2 sensors-21-06554-f002:**
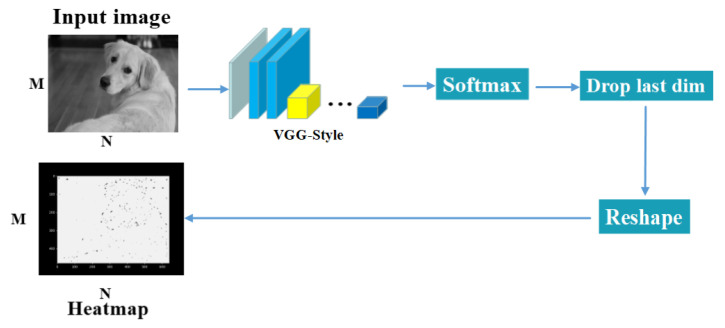
Learning point-based detection overview.

**Figure 3 sensors-21-06554-f003:**
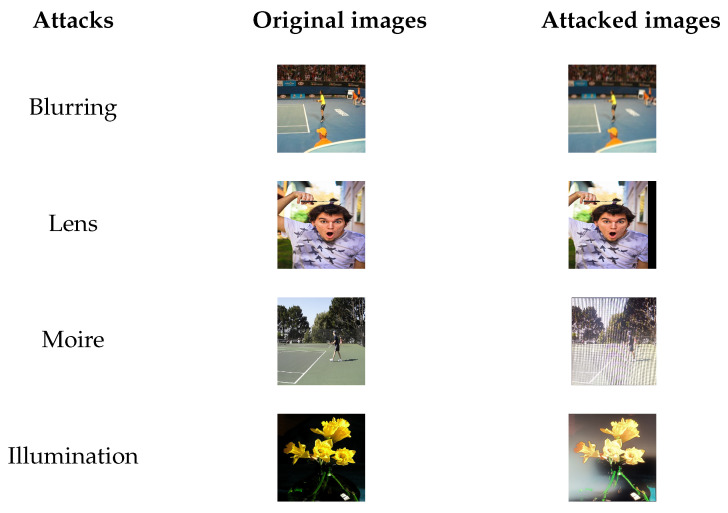
The distortion of the screen-shooting process.

**Figure 4 sensors-21-06554-f004:**
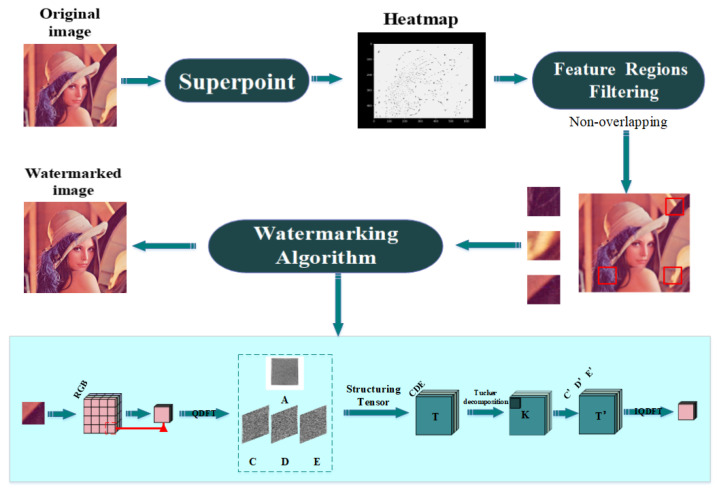
The architecture of the proposed scheme embedding process.

**Figure 5 sensors-21-06554-f005:**
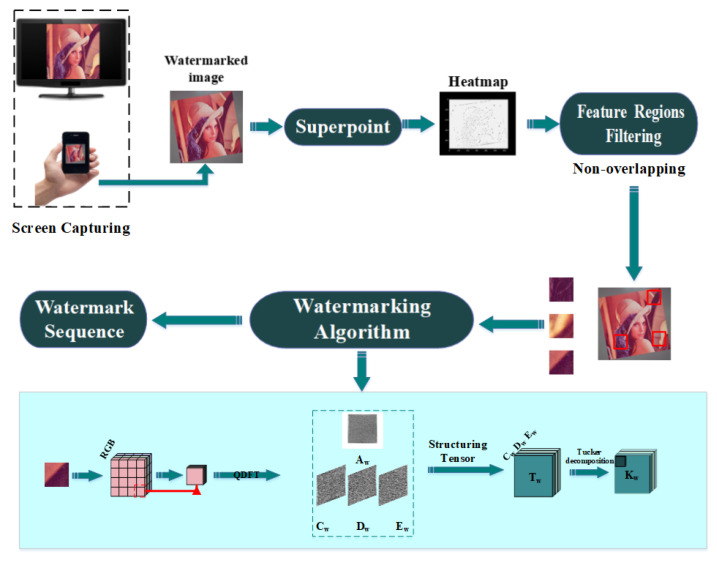
The architecture of the proposed extraction process.

**Figure 6 sensors-21-06554-f006:**
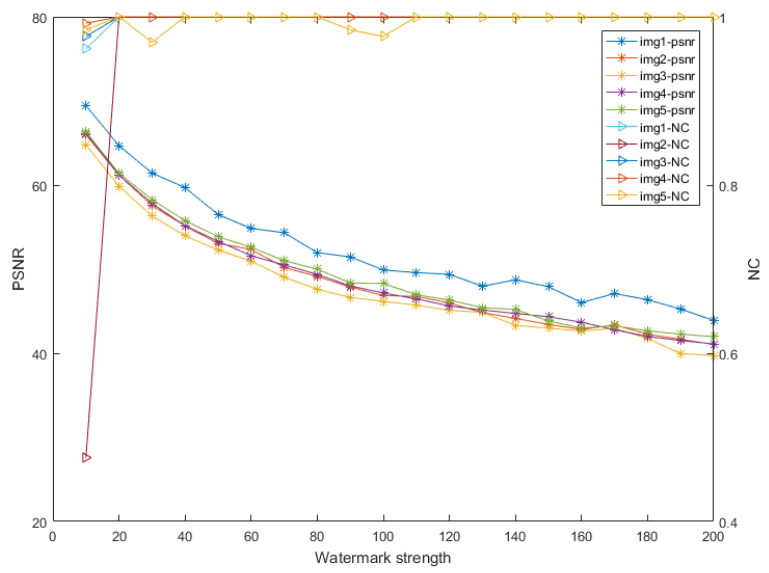
PSNR and NC of the five watermarked images with different watermark strengths.

**Figure 7 sensors-21-06554-f007:**
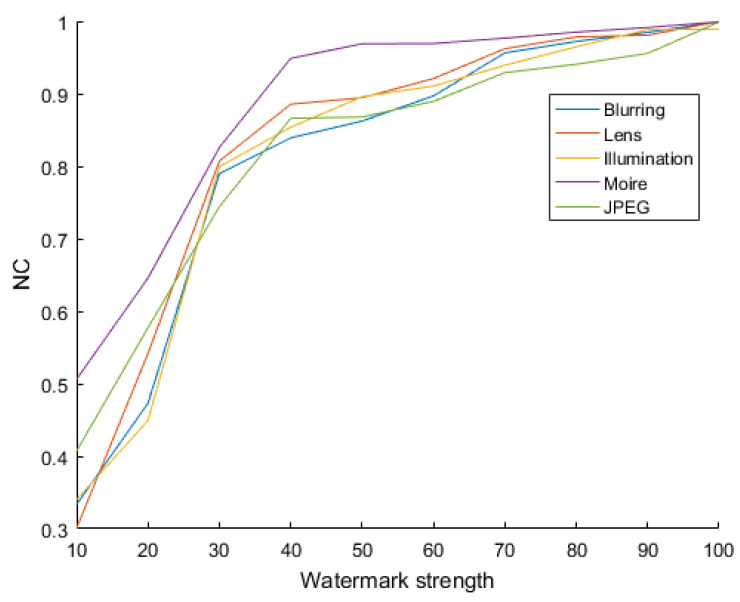
NC of the five attacks with different watermark strengths.

**Figure 8 sensors-21-06554-f008:**
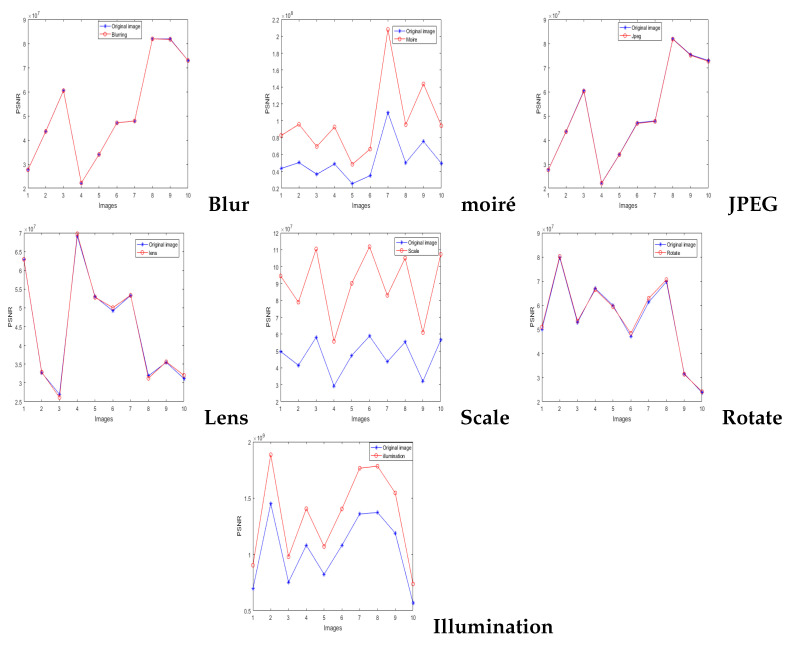
Value of core tensor K(1,1,1) resistance to different attacks.

**Figure 9 sensors-21-06554-f009:**
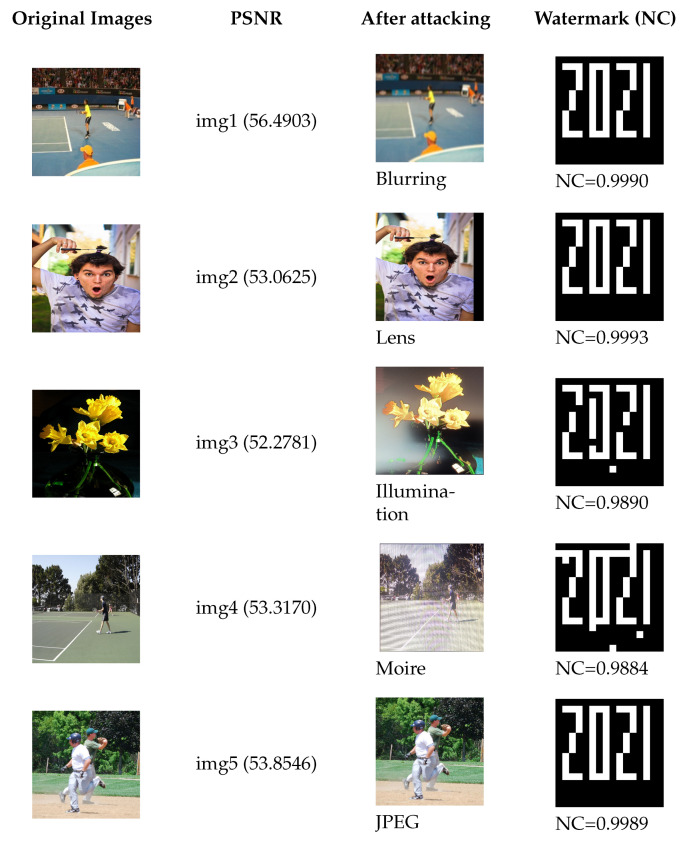
PSNR and NC values for different attacks.

**Figure 10 sensors-21-06554-f010:**
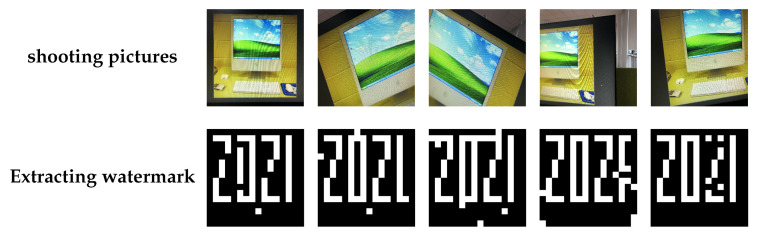
Watermark extraction performance for partial screen-shooting.

**Figure 11 sensors-21-06554-f011:**
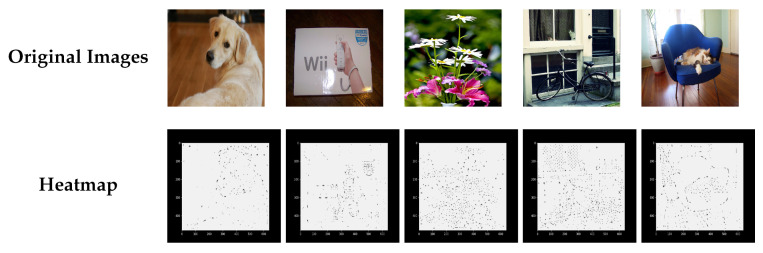
SuperPoint’s detector output probability heatmap.

**Figure 12 sensors-21-06554-f012:**
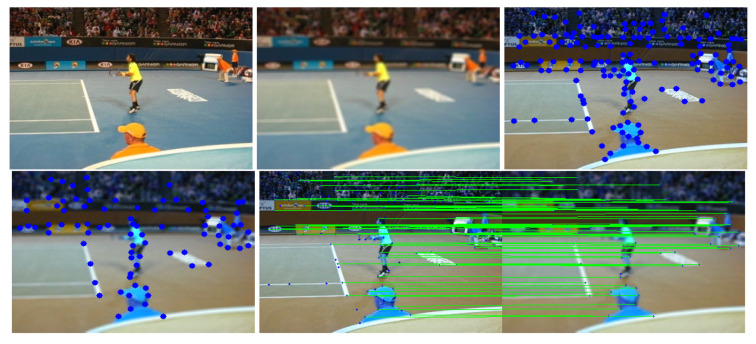
(Blurring) Row 1: Left (original image), Middle (attacked images.), Right (resulting point of original image by SuperPoint’s detector). Row 2: Left (resulting point of attacked image by SuperPoint’s detector), Right (the green lines show the correct correspondences after blurring attacks).

**Figure 13 sensors-21-06554-f013:**
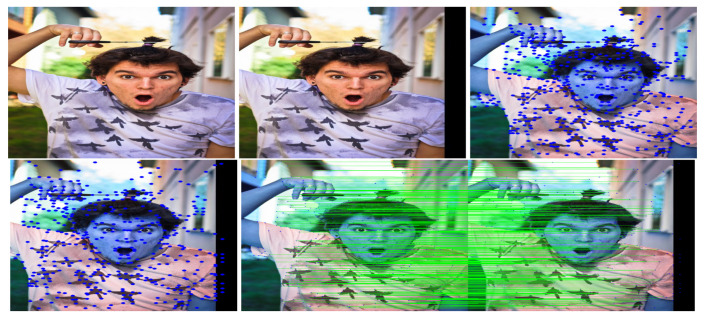
(Lens) Row 1: Left (original image), Middle (attacked images.), Right (resulting point of original image by SuperPoint’s detector). Row 2: Left (resulting point of attacked image by SuperPoint’s detector), Right (the green lines show the correct correspondences after blurring attacks).

**Figure 14 sensors-21-06554-f014:**
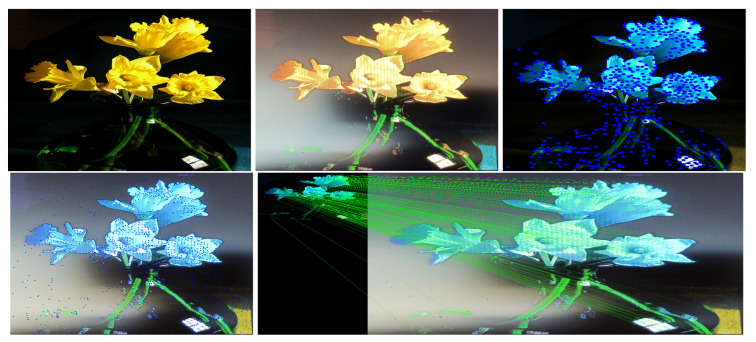
(Illumination) Row 1: Left (original image), Middle (attacked images.), Right (resulting point of original image by SuperPoint’s detector). Row 2: Left (resulting point of attacked image by SuperPoint’s detector), Right (the green lines show the correct correspondences after blurring attacks).

**Figure 15 sensors-21-06554-f015:**
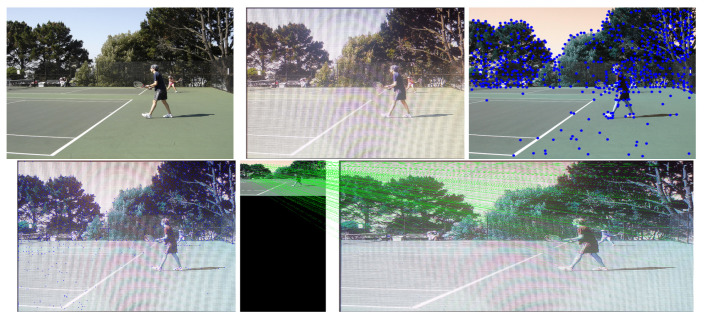
(Moire) Row 1: Left (original image), Middle (attacked images.), Right (resulting point of original image by SuperPoint’s detector). Row 2: Left (resulting point of attacked image by SuperPoint’s detector), Right (the green lines show the correct correspondences after blurring attacks).

**Figure 16 sensors-21-06554-f016:**
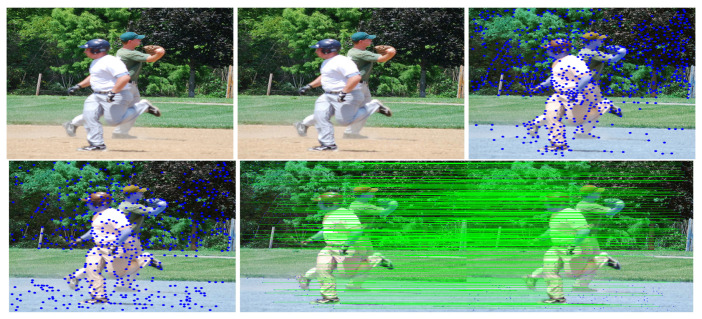
(JPEG) Row 1: Left (original image), Middle (attacked images.), Right (resulting point of original image by SuperPoint’s detector). Row 2: Left (resulting point of attacked image by SuperPoint’s detector), Right (the green lines show the correct correspondences after blurring attacks).

**Figure 17 sensors-21-06554-f017:**
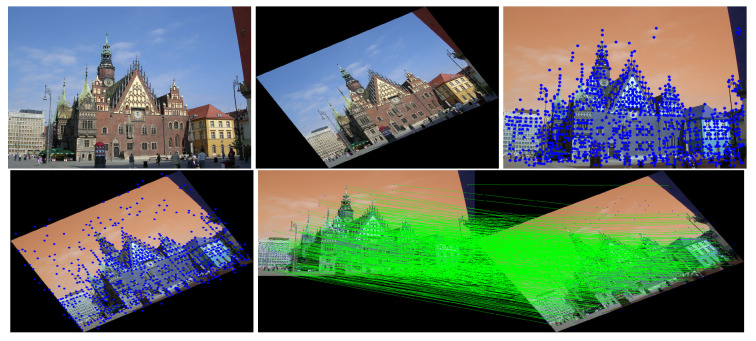
(Rotate) Row 1: Left (original image), Middle (attacked images.), Right (resulting point of original image by SuperPoint’s detector). Row 2: Left (resulting point of attacked image by SuperPoint’s detector), Right (the green lines show the correct correspondences after blurring attacks).

**Figure 18 sensors-21-06554-f018:**
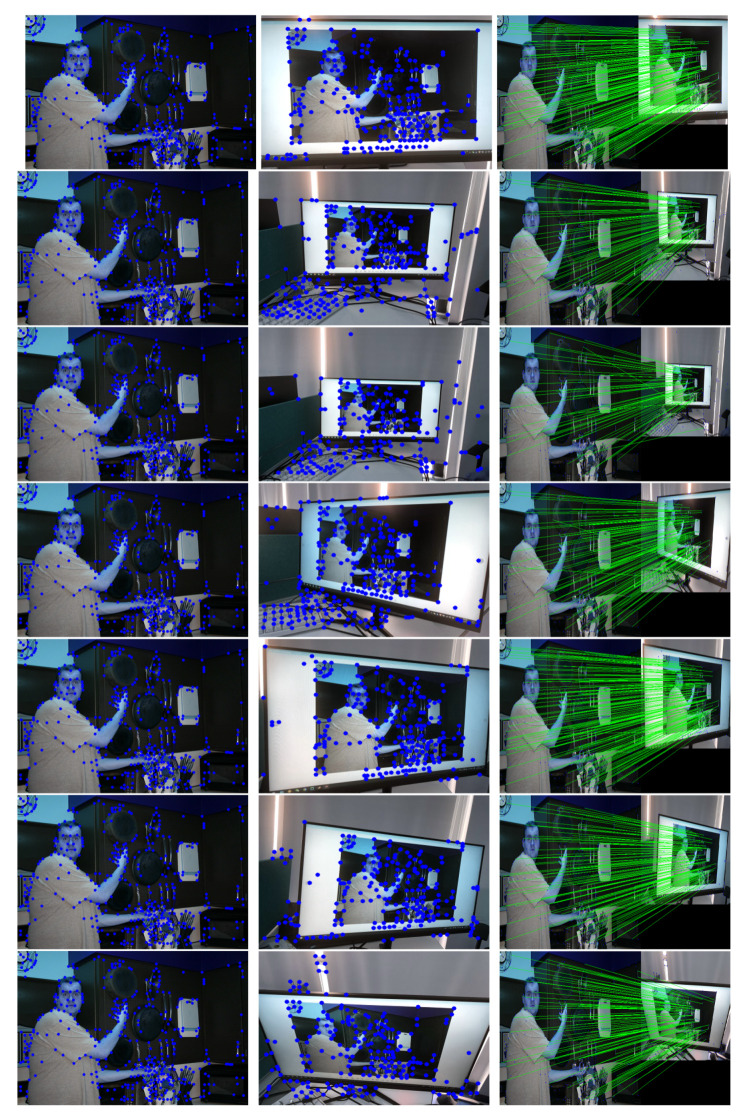
Keypoints correspondences detected with different scenarios.

**Table 1 sensors-21-06554-t001:** Perfectibility of watermarks with different strengths.

Watermark Strength	Perception Rate (25)				
	image 1	image 2	image 3	image 4	image 5
10	0/25	0/25	0/25	0/25	0/25
30	0/25	0/25	0/25	0/25	0/25
60	0/25	0/25	0/25	0/25	0/25
100	0/25	0/25	0/25	0/25	0/25
120	6/25	9/25	8/25	8/25	12/25
150	15/25	14/25	12/25	14/25	15/25
180	25/25	25/25	25/25	25/25	25/25

**Table 2 sensors-21-06554-t002:** Performance pomparison with other papers.

Schemes	Proposed Scheme	Fang [9]	Fang [17]	Cui [11]	Fang [46]	Zhang [16]
Resist Partial shooting	✔	X	X	X	X	X
Geometric Uncorrectable	YES	NO	NO	NO	NO	NO
PSNR	53.8005 dB	42.3003 dB	null	null	33 dB	null

## Data Availability

Not applicable.

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
