# Peer review of "Screen-Shooting Resilient Watermarking Scheme via Learned Invariant Keypoints and QT"

_sensors, 2021, doi:10.3390/s21196554_

Round 1

Reviewer 1 Report

The subject is very interesting and positive results are definitely worth to publish. The work is based on an earlier work of the same authors [52]; they observed that the robustness of their scheme allowed the watermark to survive even screen-shooting. For an improved presentation of the results, the following key points should be addressed.

1) It has been explained that after identifying the "feature points", a 32x32 region is chosen (disjointly) around these points, if possible. Is it true that the watermark is embedded into all of these 32x32 regions independently? If this is so, the same watermark is embedded many times, thus can be extracted many times. What about combining the extracted watermarks?

2) In the extraction procedure the "feature points" are determined again, but this set could be different form the one got by the embedding process. How and when the algorithm decides whether the region around a feature point actually contains a watermark?

3) If the above is true, then extracting the watermark is sensitive to the scaling (and affine distortion) of the image. How does this observation compare to the author's claim that no such rescaling is necessary?

4) There could be many new feature points when only a part of the protected image is visible. How those additional feature points are filtered out or handled?

5) Figures 13-19 all show the correspondence of feature points in the original and in some distorted image. I do not see the relevance of that many such images. Do the authors have an automated method to determine this correspondence?

6) All heat maps use dark-blue background and are mostly invisible. They should be recreated with white background (no feature) and grayscale to show the percentage.

7) Watermark strength, denoted by S (from Figure 7 on) has value from 0 to 200. In the description (Algorithms A and B) Step 6 has a value Q. How do these values correspond? How S is defined?

The remarks below are mainly about English. Please revise the paper for a better language.

Line 28: "which are mostly robust against ..." English
Line 29: "after" => wrong word
Line 30: "capture process" => English
Line 30: "previous" => "recent"
Line 31: ... and screnshot [8] **scenarios** have been studied extensively.
Line 32: "screen shooting **requires more sophisticated methods" English
Lines 35-38: In addition: rewrite this sentence.
Line 39: presented => proposed.
Line 40: "Existing... " => "These schemes"
Line 40: "One type is the .." => "The anti-screen ..technology is based on"
Line 43: perhaps: "... brightness of a special display pattern ..."
Line 45: "On the strength of" ==> don't use this phrase. Rather "based on" 
Line 50: "by using a ..." this sentence means that the method uses either a  camera or mobile phone. Perhaps the image is captured by one of them.
Line 56: "will" => "could" 
Line 61: "of machine learning" => "based on machine learning"
Line 61: "have exceeded" => not the methods exceeded, but their applicability, potential, etc.
Line 63: "feature points" (add s)
Line 66: "changes in season and time" => I understand it, but try to rewrite.
Line 67: "based on ... namely SuperPoint" => "based on the SuperPoint self-..."

Line 76: "cannot be" => "are not"
Line 76: "of different strengths" => delete.
Line 78: perhaps: "watermarking color images for mobile phones." Perhaps Neyman-Pearson is just a decision method somewhere, not an essential part.
Line 80: "tracking" => tracing"
Line 83: perhaps: "need to resize" => "resizing to yield an undistorted version of the original image"
Line 87: "full-map" ???? Maybe delete.
Line 89: "For our case" => "We apply a modified version of the keypoint
detector SuperPoint."
Line 94: "of a traditional" => "of traditional ... algorithms and ...      networks"
Line 96: "protecting proprietary information which is resilient to..."

Line 101: "by using" => "proposed by ... which uses a self-supervised"
Line 102 "with the same" => "of the same"
Line 104: "the model used" => "the model uses"
Line 106: "cracks" => "uses two decoders for"
Line 108" "which is a differ" => perhaps? "which depends on the architecture" 
Line 110: Figure 2, The heatmap is only a dark blue patch. Use different
colors.
Line 110: "adapt keypoint" => "adapt the keypoint" 
Line 111: "detector are" => "detector is"
Line 111: "exactly" => delete
Line 112: "which requires" => "and requires"
Line 117: "Sangwine [50] was the first" => "We take the description of QDFT from Sangwine [50]." 
Line above (3): "is defined by" => "is" (this IS the inverse, you do not define it).
Line 121: "is any unit" => "can be any unit"
Line 123: "paper we take" (insert "we take")
Line 123+1: "which is a" => "the", "which are the" => "the".
Line 124: C,D,E,T should be italics

Line 128: "can be considered" => "are considered"
Line 131: "...decomposition, three" => "by the Tucker decomposition to three"
Line 132: "are obtained [48]" => "[48]"

Line 148: "merits of a ... algorithm and a ... network" => "merits of ... algorithms and ... networks"
Line 153: "We then describe ... model as follows" => "We first describe ... model in Subsection 3.1"
Line 162: "should not overlap, the" 
Line 162: "should be shifted. The shift operation"
Lines 165-168: Rewrite. Perhaps it is enough to say that R(a_k) are regions of size 32x32 which should be disjoint. DESC is not defined.
Line 169: "may cause" => "causes clustering"
Line 170: "more" => "additional"
Line 181: "Locate high confidence keypoints I_h and obtain the coordinate set S of the keypoints".
Line 184: "with a size of" => "of size"
Line 187: "The main target of the proposed QDFT and TD watermarking scheme in [52] was applications in normal image attack scenario."
Line 189: "is really robust" => "turned out to be robust"
Line 193: "correlations" => "correlation"
Line 200: "bring" => "brings"
Line 201: "they can" => "it can" ???
Line 219: "a coordinates set" => "the coordinate set"
Figures 4-5: the heatmap should not be dark blue

Line 243: "select" => "selected"
Line 266: "the 10 image" What are those ten images?
Figures 6 and 11: Figure 11 should be compressed, and to refer to Figure 6
Line 286: "We lead" => "We designed"
Line 187: "study is" => "study was"
Line 295: Perhaps students do not see the watermarks, but judge the "presence of watermarks" Please rewrite accordingly
Line 296: "by any" => "by our ... participants"
Line 300: "that is, detection"
Line 301: "more bluer" => "darker blue"
Line 303: "SuperPoint output as probability heatmap."
Figure 12 (and other figures with heatmaps) I suggest inverting the heatmaps to have white background
Line 307: English. 
Line 309: I don't think you want to embed multiple regions. Please explain
Line 312: "provide the" => "provide some" "result" => "results"

Author Response

I should like to express my appreciation to you and the referees for suggesting how to improve our paper and hope that we have now produced a more balance and better account of our work. You are one of the dedicated people I have ever met. Finally, sincerely thank you for reviewing my manuscript.

Reviewer 2 Report

The work submitted for review is interesting.

The work contains a lot of information which makes it seem quite "heavy".

The methods are described in detail - but as far as the material is concerned, I miss information about the origin of the photos, their quality, number, etc. Please complete this information. 

I think it is also worth supplementing the abstract section with information related to the results and it is worth expanding the conclusion section.

I think the rest of the work is fine.

Author Response

(The authors gave the same response as above.)

Reviewer 3 Report

The paper presents a screen-shooting watermarking scheme via learned invariant  keypoints. In particular, the proposed scheme combines FRFS (Feature regions filtering model to superpoint), QDFT (Quaternion discrete Fourier transform), and FRFSQT (Feature regions filtering model to superpoint, quaternion discrete Fourier transform, and Tensor decomposition) for protecting confidential information displayed on the screen.

The paper sounds good.....however "Introduction" should be shortened, and considerations about previous experiences should be moved to "related work" section.

Related work section should not be restricted to the arguments at the basis of the proposed scheme.

In Section 2.2 formulas should be better explained ...

Author Response

(The authors gave the same response as above.)

Round 2

Reviewer 1 Report

The authors' response cleared many of my concerns. It is a pity that not all the explanations got into the final version. 

I understand that the software produces the blue-red heatmap. Nevertheless it would be much better to convert it to grayscale. As an example I did it for one of the maps (see the attached pdf file). The authors could do the same which would definitely improve the quality of the paper.

Author Response

I have read the referee’s comments very carefully and download the attached pdf file. The previous round review, I apologize for failing to grasp your comments.

In addition, I convert the heatmap to grayscale. Referee’s comments definitely improve the quality of our manuscript.
